# Repositioned Drugs for Chagas Disease Unveiled via Structure-Based Drug Repositioning

**DOI:** 10.3390/ijms21228809

**Published:** 2020-11-20

**Authors:** Melissa F. Adasme, Sarah Naomi Bolz, Lauren Adelmann, Sebastian Salentin, V. Joachim Haupt, Adriana Moreno-Rodríguez, Benjamín Nogueda-Torres, Verónica Castillo-Campos, Lilián Yepez-Mulia, José A. De Fuentes-Vicente, Gildardo Rivera, Michael Schroeder

**Affiliations:** 1Biotechnology Center (BIOTEC), Technische Universität Dresden, 01307 Dresden, Germany; melissa.adasme@tu-dresden.de (M.F.A.); sarah_naomi.bolz@tu-dresden.de (S.N.B.); lauren.adelmann@mailbox.tu-dresden.de (L.A.); sebastian.salentin@tu-dresden.de (S.S.); joachim.haupt@tu-dresden.de (V.J.H.); 2Facultad de Ciencias Químicas, Universidad Autónoma Benito Juárez, Oaxaca 68120, Mexico; arimor10@hotmail.com; 3Departamento de Parasitología, Escuela Nacional de Ciencias Biológicas, Instituto Politécnico Nacional, Ciudad de México 11340, Mexico; bnogueda@yahoo.com (B.N.-T.); shigella715@hotmail.com (V.C.-C.); 4Unidad de Investigación Médica en Enfermedades Infecciosas y Parasitarias, UMAE Hospital de Pediatría, Centro Médico Siglo XXI, Instituto Mexicano del Seguro Social, México City 06720, Mexico; lilianyepez@yahoo.com; 5Laboratorio de Investigación y Diagnóstico Molecular, Instituto de Ciencias Biológicas, Universidad de Ciencias y Artes de Chiapas. Libramiento Norte Poniente 1150, Col. Lajas Maciel, Tuxtla Gutiérrez 29029, Mexico; jose.defuentes@unicach.mx; 6Laboratorio de Biotecnología Farmacéutica, Centro de Biotecnología Genómica, Instituto Politécnico Nacional, Reynosa 88710, Mexico; gildardors@hotmail.com

**Keywords:** Chagas, *Trypanosoma cruzi*, drug repositioning, structural screening, non-covalent interactions

## Abstract

Chagas disease, caused by the parasite *Trypanosoma cruzi*, affects millions of people in South America. The current treatments are limited, have severe side effects, and are only partially effective. Drug repositioning, defined as finding new indications for already approved drugs, has the potential to provide new therapeutic options for Chagas. In this work, we conducted a structure-based drug repositioning approach with over 130,000 3D protein structures to identify drugs that bind therapeutic Chagas targets and thus represent potential new Chagas treatments. The screening yielded over 500 molecules as hits, out of which 38 drugs were prioritized following a rigorous filtering process. About half of the latter were already known to have trypanocidal activity, while the others are novel to Chagas disease. Three of the new drug candidates—ciprofloxacin, naproxen, and folic acid—showed a growth inhibitory activity in the micromolar range when tested ex vivo on *T. cruzi* trypomastigotes, validating the prediction. We show that our drug repositioning approach is able to pinpoint relevant drug candidates at a fraction of the time and cost of a conventional screening. Furthermore, our results demonstrate the power and potential of structure-based drug repositioning in the context of neglected tropical diseases where the pharmaceutical industry has little financial interest in the development of new drugs.

## 1. Introduction

Chagas disease, also known as American trypanosomiasis, is a life-threatening infection caused by the protozoan parasite *Trypanosoma cruzi* [1]. According to the World Health Organization (WHO), about 7,000,000 people worldwide are afflicted with Chagas disease. Most cases of Chagas disease occur in South America, mostly in rural areas, with the infection being endemic in 21 countries [2]. The parasite is often transmitted via Triatomine insects. The infection occurs when the insect bites and subsequently defecates in the bite, allowing *T. cruzi* to enter the bloodstream. Moreover, *T. cruzi* can be congenitally transmitted from a pregnant mother to her baby, via blood transfusion, organ transplantation, or even due to laboratory accidents [1].

The acute phase of the disease occurs during the first few months after infection. *T. cruzi* propagates in the bloodstream, which produces mild symptoms, such as a skin lesion at the infection site, headache, fever, and muscle aches [3]. During the chronic phase of infection, the parasite lodges itself mainly in digestive and cardiac tissues. During this phase, about 30% of patients suffer from cardiac issues and 10% suffer from digestive or neurological issues, which can be fatal [2]. Fatality is frequently caused by Chronic Chagas Cardiomyopathy, which is the weakening of the heart muscles due to the parasite invasion.

Currently, there are only two drugs on the market for Chagas disease: Benznidazole works via inducing reductive stress, whereas Nifurtimox causes the generation of free radicals. Both drugs cause the parasite to be vanquished within 60–90 days. However, they are only effective in the predominantly asymptotic acute phase of the disease [4]. Once the disease reaches the chronic stage, there is not much that can be done. Furthermore, both drugs produce severe side effects in over 40% of patients and are contraindicated for use in pregnancy, reducing their applicability. Nifurtimox has severe side effects related to the nervous system, including depression, anorexia, neuropathy, insomnia, headache, and vomiting. On the other hand, Benznidazole has severe toxicities related to skin hypersensitivity, such as dermatitis and severe symptoms like depression of bone marrow, thrombocytopenic purpura, and agranulocytosis [5]. Due to the unspecific mechanism of action, the severe side effects, and the limited efficacy of the current chemotherapeutic options, there is a need for improved drugs with targeted action and less severe side effects.

The cost for pharmaceutical companies to research, develop, test, and bring a new drug to market is about $2.6 billion and takes about 10–15 years [6]. Drug repositioning, sometimes referred to as drug repurposing, is the utilization of already approved or experimental drugs for a novel indication [7]. The risks and development costs are reduced as there is already a wealth of knowledge available for approved and experimental drugs, such as safety, adsorption, distribution, metabolism, excretion, and other biological data, as well as clinical data in some cases [7]. In fact, about 60% of all drugs, both approved and experimental, have been tested for more than one disease [8]. The need for more effective and less toxic drugs coupled with the low commercial interest of pharmaceutical companies makes Chagas disease the perfect case for drug repurposing.

Several studies have reported repositioning candidates for Chagas with promising trypanocidal effects. Examples are Amiodarone, which is actually used as a Class III anti-arrhythmic agent [9]; Trimetrexate, an antifolate drug used against *Pneumocystis carinii* infection in patients with Acquired Immune Deficiency Syndrome (AIDS) [10]; and, most relevant, Posaconazole and Ravuconazole, which entered phase II clinical trials. Unfortunately, the latter showed poor results compared to Benznidazole [11,12]. Still, combination therapy could lead to better results [13]. Nowadays, with the exponential growth of structural data, it is possible to exploit drug repositioning at a structural level and to screen vast amounts of drug–target interactions to predict polypharmacological potential and repositioning opportunities [14]. For instance, Haupt et al. explored shared binding sites between Chagas targets and other proteins to identify novel drugs for the treatment of Chagas disease. Using their approach known as Target Hopping, they predicted that the antiviral Foscarnet would inhibit the target Farnesyl Pyrophosphate Synthase (FPPS) in *T. cruzi* [15]. In a more recent study, a virtual screening approach combining classical docking with protein–ligand interaction profiling identified drug repositioning candidates against *T. cruzi* infection. Nilotinib, Glipizide, Glyburide, and Gliquidone were predicted to bind to the Chagas target Dihydrofolate Reductase-Thymidylate Synthase (TcDHFR-TS) with high affinity. They were tested on *T. cruzi* epimastigotes, where they showed a growth inhibitory activity in the micromolar range, making them potential lead compounds in the development of new treatments for Chagas disease [16].

Over time, multiple *T. cruzi* enzymes have been highlighted as crucial therapeutic targets [15,17,18]. With the aim to identify novel repositioning candidates for Chagas disease, we explored the wide space of FDA approved drugs using a novel hybrid structure/knowledge-based repositioning method presented in this work.

## 2. Results

### 2.1. Identification of Relevant Chagas Targets and Their Structural Data

In the case of Chagas disease, a “good” target is a protein that is found in *T. cruzi* and not in humans and is important to parasite survival, or, alternatively, a *T. cruzi* protein that is different enough from its human homolog to reduce off target effects. It is important to consider that the current chemotherapeutic options for Chagas—nifurtimox and benznidazole—do not act through a primary target but kill the parasite by generating free radicals and inducing reductive stress, respectively [4]. To find a drug with a more focused mechanism of action, a list of drugable targets first had to be established.

As the starting point of this work, a set of 20 Chagas targets was assessed. Structural data was available for 16 targets (see Table 1). The targets were evaluated based on their dependability as a Chagas target. The seven targets listed above the bold line in Table 1 have been thoroughly researched and there is a high confidence that their modulation will produce the desired effect in *T. cruzi*, while the nine targets below the line have at least been researched as *T. cruzi* targets.

As for any other structural approach, the availability of 3D structural data of proteins and ligands is a must. Currently, the Protein Data Bank (PDB) holds more than 169,000 structures, with a current annual growth of nearly 12,000 entries [32]. It is estimated to cover the vast majority of the known drug targets (about 92%) [33], and most of the structures (more than 60%) are in complex with biologically relevant ligands [34]. Most of the Chagas targets have at least one structure with a biologically relevant ligand available in the PDB (see Table 1), with the exception of cyclic nucleotide specific phosphodiesterase, Dihydrolipyl dehydrogenase, Ribose-5-phosphate dehydrogenase, and Triosephosphate Isomerase. Interestingly, only five structures of dihydroorate contained relevant ligands, although there were 58 structures available. The other complex structures involved cofactors, molecules from the buffer solution, or molecular fragments.

### 2.2. In Silico Screening

We analyzed drug interaction pattern similarities between binding sites of the query complexes and over 130,000 protein structures. The identified targets comprised 22 binding sites in complex with several relevant ligands (Table 1). Each binding site accounted for an independent screening. The screening output was a list of hit complexes with high interaction pattern similarity ranked by *p*-value and aggregated by ligand. In total, there were 523 hits across the 22 screenings, ranging from 0 to 97 hits per screening with a mean of 23. There was a modest positive correlation between the number of query structures and the number of hits, as indicated by a Pearson correlation coefficient of 0.6. Several compounds yielded a hit in multiple screenings, while others were unique to only one screening. Most of the frequent hitters were not relevant to Chagas disease, such as glutathione, citric acid, and different amino acids. On the other hand, some screenings produced hit numbers well above the average, such as the screenings of squalene synthase with 96 hits and lanosterol demethylase with 69 hits.

To get a first impression of the significance of the screening predictions, we checked the Binding DB [35] for binding evidence. Affinity data were available for only three drug target pairs. Of these three, the binding of Risedronate to the target FPPS was also predicted by our screening. In contrast, there was binding evidence for Etravirine and the target Cruzipain as well as Chlorpromazine and Trypanthione reductase, but these drug target pairs were not a hit in the screenings.

Following several filtering steps (see Methods section), the primary list of 512 compounds was narrowed down to 38 high-priority predictions (see Table 2). These compounds were selected as our top repositioning candidates for Chagas disease.

### 2.3. Characterization of the Hit Candidates

The 38 hits cover a diverse set of primary indications (see Table 2), such as osteoporosis, cancer, asthma, antifungal, antibiotics, and antivirals, among others. Surprisingly, Pyrimethamine and Quinine are the only two drugs with a parasitic effect (Toxoplasmosis and Malaria, respectively) as primary indication. Independently of their primary indication, the hit candidates were classified according to their novelty for Chagas. According to literature evidence, 20 drugs have a direct evidence of Chagas activity(++), five drugs have rather an indirect link to Chagas(+), and the other 13 have no previous evidence(?), meaning they are fully novel to Chagas disease.

Moreover, the chemical space of the selected hit drugs was characterized to determine the scaffold diversity that the screening achieved. As shown in Figure 1, overall there is very little chemical similarity between the top hits, with only 1.5% of the similarity scores above the threshold of 0.7. In general, most hits have unique chemical profiles and therefore remain without cluster. Nonetheless, a few clusters can be spot in the similarity chart and are worth further exploration. For instance, there is a small cluster (cluster B) with the highest chemical similarity score (0.87) grouping the adamantane derivative rimantadine and memantine. Cluster C groups compounds derivative of folate, e.g., folic acid, pemetrexed, and leucovorin, with a mean similarity of 0.77 ± 0.29. In cluster A, there is a big group of nucleic acids and their analogs with a relatively high mean similarity of 0.62 ± 0.21. Although they all share similar core scaffolds, the rest of their chemical structure is very distinct from one another. Finally, the chart shows no clear relation between the chemical structure of the hits and the novelty of their trypanocidal activity (color tags along the left side of the chart). The hits cover a vast chemical space, regardless of whether they have a direct (green), indirect (orange), or no (red) trypanocidal evidence.

### 2.4. Inhibitory Effect of Selected Drugs in *T. cruzi* Epimastigotes and Trypomastigotes

From the list of high priority drug repositioning candidates, eight were selected for further experimental validation. The selection was based on the following criteria; novelty of Chagas activity (+ or ?), a reasonable interaction pattern similarity between query and hit (positive visual inspection), and availability at a low price.

The trypanocidal activity of the eight candidates was assessed in vitro for the epimastigote stage and ex vivo for the trypomastigote stage using the *T. cruzi* strains Ninoa and INC-5 (see Table 3). Of the compounds tested, ciprofloxacin, folic acid, and naproxen showed the greatest inhibitory activity against *T. cruzi* trypomastigotes, with IC_50_ values of 25.7 µM, 28.1 µM, and 58.5 µM for the Ninoa strain, and 21.3 µM, 21.5 µM, and 38.3 µM for the INC-5 strain, respectively. Moreover, the active drugs demonstrated relatively low cytotoxicity with CC_50_ (the concentration of the drug that causes the death of 50% of viable cells in the host) values of 7.7 × 10^23^ µM for ciprofloxacin, 2.5 × 10^18^ µM for naproxen, and 9.1 × 10^17^ µM for folic acid. Based on these results, the selectivity indices (CC_50_/IC_50_) were 2.9 × 10^22^, 4.2 × 10^16^, and 3.2 × 10^16^, respectively. While ciprofloxacin, folic acid, and naproxen had IC_50_ values in the same range as the standard Chagas disease medications nifurtimox and benznidazole, they exhibited lower cytotoxicity and higher selectivity in comparison. Yet, none of the drug repositioning candidates showed trypanocidal activity in the lower micromolar range against *T. cruzi* epimastigotes (see Table 3).

### 2.5. In Vivo Trypanocidal Activity of the Drug Repositioning Candidates

As ciprofloxacin, naproxen, and folic acid showed a growth inhibitory activity in the micromolar range against *T. cruzi*, they were tested for parasitemia inhibition in vivo. Mice were infected with blood trypomastigotes and at day 13 post-infection, a single dose (at 100 mg/kg body weight) of the drugs ciprofloxacin, nifurtimox, or folic acid was orally administered. Parasitemia was monitored at 2, 4, 6, and 8 h after administration (see Figure 2).

All drugs tested demonstrated inhibition of parasitemia. Eight hours after administration to the mice infected with the Ninoa strain of *T. cruzi*, folic acid, ciprofloxacin, and naproxen showed parasitemia inhibition of 57.2%, 66.7%, and 85.8%, respectively, while the established Chagas treatment nifurtimox exhibited parasitemia inhibition of 77.8%. Interestingly, the trypanocidal activity of naproxen was significanclty higher than that shown by nifurtimox (*p* ≤ 0.05). In mice infected with the *T. cruzi* strain INC-5, folic acid, ciprofloxacin, and naproxen showed parasitemia inhibition of 66.7%, 43.8%, and 42.9%, respectively, eight hours after administration. In comparison, nifurtimox exhibited an inhibition of 72.3%. It is worth mentioning that folic acid showed significantly higher activity than nifurtimox against INC-5 strain at 4 and 6 h after drug administration (*p*≤ 0.05), although it was maintained at 8 h, it was lower than the reference drug. Overall, the drug repositioning candidates achieved parasitemia inhibition of *T. cruzi* trypomastigotes in vivo.

### 2.6. Characterization of the Validated Drugs

Ciprofloxacin, naproxen, and folic acid were identified as drug repositioning candidates for Chagas disease based on the tanimoto similarity score of the interaction fingerprints between the query and the hit complexes of the screening. In other terms, the screening identified a high non-covalent interaction pattern similarity between the drugs in complex with their original targets and the therapeutic Chagas targets in complex with their original binders. Figure 3A shows the query complex of the Chagas target trans-sialidase (sialic acid site) and its original binder deoxy-N-acetylneuraminic acid. Among the interactions that define the binding mode of the query, there is a clear set of interactions (a pattern) that are also present in the hit complex of ciprofloxacin and its original target outer membrane porin F. The pattern includes a pair of salt bridges (1), three hydrogen bonds (2), and one hydrophobic interaction within the same distance and angle.

The same applies to the repositioning of naproxen (see Figure 3B), where there is a common interaction pattern between FPPS (homoallylic site) with its binder zoledronate (query) and naproxen with its original target serum albumin (hit). In the case of folic acid, there is an explicit interaction pattern that is present both in the complex of TcDHFR with its binder methotrexate (query) and in the complex of folic acid with human DHFR (hit) (see Figure 3C). However, it should be noted that the protein targets are homologs and the residues that define the binding pocket are very similar.

## 3. Discussion

### 3.1. The Diversity of Chagas Targets and the Promiscuous Hits

Over the years, several *T. cruzi* enzymes have been identified as therapeutic targets (Table 1) and they cover different enzyme classes like hydrolases, isomerases, transferases, and oxidoreductases, with the majority of them being either transferases or oxidoreductases. Consistent with this tendency, the PDB mainly contains hydrolases, transferases, and oxidoreductases, which currently make up 38.9%, 30.4%, and 15.7% of the enzyme structures, respectively [32].

The binding of a molecule to two or more different targets is known as drug promiscuity [56,57]. In drug development, compound promiscuity is often regarded as an unfavorable property because drugs binding multiple targets are expected to cause side effects. However, promiscuity can also positively affect the efficacy of a drug, i.e., a drug binding several targets linked to the same disease will have an increased impact compared to a single-target modulator. Before prioritization of the screening results, several hit drugs were predicted to bind multiple targets. More surprisingly, some multi-target hits, such as amino acids, glycerol, glutathione, citric acid, and bile salt, did not even discriminate on enzyme class. There is a significant correlation between compound promiscuity and binding site similarity [58]. According to Gao et al., the number of ways a drug can interact with its target is limited to 1000 distinct interface types [59]. Therefore, the promiscuous binding of the hit drugs to several targets is not an exception but rather a frequent observation. The multi-target hits could also be directly related to the coincidence of targets among the same pathways (Appendix A). For instance, FPPS, lanosterol demethylase, and squalene synthase are all involved in ergosterol biosynthesis; DHFR and pteridine reductase are involved in folate metabolism; trypanothione reductase and spermidine synthase are involved in trypanothione metabolism; and finally, GADPH, G6PDH, and triosephosphate isomerase are involved in glycolysis.

Another reason for the promiscuous predictions might be unspecific and simple interaction patterns. For instance, it is known that hydrogen bonds and hydrophobic contacts occur more frequently than other interaction types and therefore lead to less specific predictions [60]. Many of the multi-target hits are very important substances for metabolism and other cellular functions [61,62]. As these ligands are involved in multiple pathways and bind several different enzymes, it makes sense that simple interactions are preferred.

### 3.2. Predicted Drugs with a Known Activity in Chagas Disease

More than half of the drugs predicted by the drug repositioning screening had a previous evidence of trypanocidal activity (Table 2). They serve as a good validation of the approach and demonstrate its ability to retrieve drugs with a known link to the disease. Nonetheless, the quality of evidence varies from case to case.

Risedronate, for example, is an osteoporosis medication that acts by inhibiting poly-isoprenoid biosynthesis. It has been shown to bind *T. cruzi* FPPS [41] and its trypanocidal activity against *T. cruzi* has been confirmed [63,64].

Pyrimethamine, on the other hand, has a defined trypanocidal activity with a proposed mechanism of action via the target predicted by the screening but in a different organism. It is an antiparasitic used to treat malaria by inhibiting the target DHFR, which is an important enzyme in folate metabolism and the production of the nucleic acid thymidine. The disruption of thymidine biosynthesis consequently interferes with DNA biosynthesis [19]. DHFR is also a well-known Chagas target and essential for the survival of *T. cruzi*. Several studies have already explored pyrimethamine in the context of trypanosoma diseases, African sleeping sickness (caused by *Trypanosoma brucei*), and Chagas disease. In fact, pyrimethamine has been shown to be a potent inhibitor of *T. cruzi* DHFR with a K_i_ of 0.1
μm [19] and has been reported to inhibit *T. cruzi* growth in vitro with an EC_50_ of 3800 nm [13].

Another example is Zidovudine, an antiretroviral used in the treatment of HIV [65], which has an anti-trypanosomal activity known from phenotypic studies. Yet, the target and mechanism of action remain unknown. Nakajima et al. reported a growth inhibition of *T. cruzi* by zidovudine with an $IC_{50}$ between 200 and 300 nm. However, they were unable to elucidate the mechanism of the trypanocidal activity. While *T. cruzi* has genes for reverse transcriptase, which are activated in several stages of its life cycle, zidovudine does not inhibit *T. cruzi* reverse transcriptase as it was initially thought [44]. This suggests that the activity of zidovudine relies on the inhibition of another enzyme. The prediction made in our screening proposes for the first time a mechanism of action for the trypanocidal activity of zidovudine. It was predicted to bind Squalene synthase (SQS) (see Table 2), which is relevant to T. cruzi survival due to its role in Ergosterol synthesis. Ergosterol is an essential membrane sterol in many trypanosomatid parasites that plays the same structural role as cholesterol in humans [17].

At a lower level, there are drugs with a weak evidence of indirect trypanocidal activity. For instance, Progesterone is a female steroidal hormone important in pregnancy with various medical applications, e.g., birth control, fertility treatments, and hormone replacement therapy [66]. It has not been shown to kill *T. cruzi* in vitro or in vivo, but there is indirect evidence of its potential effectiveness. It has been reported that female infected mice with high endogenous progesterone levels also show low parasite loads [36].

### 3.3. Predicted Drugs with a Novel Activity to Chagas

Three predicted drugs with no previous evidence of activity in Chagas disease were validated as inhibitors of *T. cruzi* proliferation: ciprofloxacin (a “normal” antibiotic), naproxen (an over the counter pain killer), and folic acid (a vitamin).

Ciprofloxacin is a broad-spectrum antibiotic of the semisynthetic fluoroquinolone class. Fluoroquinolones have two main bacterial targets: the type II topoisomerases DNA gyrase and DNA topoisomerase IV [67]. Type II topoisomerases introduce a transient double-strand break into DNA segments, pass an intact double strand through this break, and re-ligate the break to relax topological stress or generate supercoils [68]. Fluoroquinolones inhibit the ligase activity of bacterial type II topoisomerases, thereby blocking bacterial growth [69]. Several studies have reported the relevance of fluoroquinolones as parasitic inhibitors and suggested inhibition of DNA topoisomerases as mechanism of action [38,70,71,72]. In our screening, ciprofloxacin was predicted to bind trans-sialidase (see Figure 3), which catalyses the transfer of sialic acid from host glycoconjugates to acceptor molecules on the parasite surface and is fundamental for *T. cruzi* survival [73]. The prediction suggests for the first time that the antiparasitic effect of quinolones is caused by trans-sialidase inhibition. Moreover, Nenortas et al. have reported that ciprofloxacin is active against *T. brucei* in vitro with an $EC_{50}$ of 52 μm [70]. We have now shown that ciprofloxacin is also a potent inhibitor of *T. cruzi* trypomastigotes with an $IC_{50}$ of 21 μm (see Table 3) and blocks *T. cruzi* growth in vivo. Generally, ciprofloxacin is considered a relatively safe, well tolerated, and widely used drug in clinical practice. As for most other drugs on the market, several side effects have been reported that among others cause gastrointestinal, central nervous system, skin, cardiovascular, lymphatic, or nutritional issues. However, most of them are reversible and occur at a mild or moderate intensity (in about 94% of cases) [74]. The widespread use of ciprofloxacin has caused a remarkable increase in bacterial resistance to the drug, making it a less effective antibiotic [75]. For this reason, its repositioning to a parasitic indication would be sustainable and profitable

Naproxen is a bicyclic propionic acid derivative that has analgesic, anti-inflammatory, and antipyretic effects and is classified as non-steroidal anti-inflammatory drug (NSAID). Like most NSAIDs, naproxen acts via the inhibition of Cyclooxygenase (COX) isoforms I and II, which are involved in the synthesis of prostaglandins, prostacyclin, and thromboxane from arachidonic acid [76,77]. Our results show that naproxen is a potent inhibitor of *T. cruzi* trypomastigote (see Table 3). Furthermore, our screening predicted that naproxen binds *T. cruzi* farnesyl pyrophosphate synthase (TcFPPS) (see Figure 3). TcFPPS synthesises farnesyl pyrophosphate (FPP). Farnesylation is a post-translational modification required for membrane localization of many proteins, such as small GTPases. Therefore, the inhibition on TcFPPS disrupts various cellular functions [41]. This makes TcFPPS a well-known Chagas target and could explain the trypanocidal activity of naproxen. Although naproxen effectively inhibits *T. cruzi* proliferation, several aspects should be considered prior to its repositioning. The safety of NSAIDs with regards to cardiovascular events has been evaluated in a large number of retrospective and prospective clinical studies, all of which have reported cardiotoxicity [78]. Moreover, Cossentini et al. demonstrated that the gastric mucosal damage caused by the intake of Aspirin, a commonly used NSAID, facilitates *T. cruzi* infection by the oral route [79]. While this has not been tested particularly for naproxen, the side effects in the gastrointestinal tract and kidneys are a well-known disadvantage of NSAIDs [78]. Additionally, a comparison of the structures of trypanosomal and human FPPS revealed that their active site residues are highly conserved, which makes the design of parasite-specific drugs difficult [41].

Folic acid is the manufactured version of folate. As the human body is not able to produce folate on its own, folic acid is used as a dietary supplement and food fortification. In our screening, folic acid was predicted to bind TcDHFR. The repositioning has been proposed between the same protein among different species (from human to *T. cruzi* DHFR), meaning that the targets are expected to have similar binding sites and thus form similar non-covalent interactions with their ligands. Although the repositioning itself is not very surprising, we demonstrated the trypanocidal activity of folic acid in this study for the first time. However, there is no clear explanation for the trypanocidal activity of folic acid since it is a known substrate of the predicted target. Several studies have discussed the positive effect of folic acid intake in different diseases and conditions. Folate plays an essential role in the human body as a major coenzyme in one-carbon metabolism, including DNA synthesis (dTMP) and methylation [80]. Moreover, folate deficiency has been associated with a risk for several diseases, involving, among others [81], cancer [82,83,84], Alzheimer [85], hypertension [86], and some pregnancy and birth complications, such as megaloblastic anemia [87] or neural tube defects (NTDs) [88,89]. Moreover, it has been shown that the cell-mediated immunity is highly affected by folate deficiency [90]. In fact, the blastogenic response of T lymphocytes to certain mitogens is decreased in folate-deficient humans and animals, and the thymus is preferentially altered. In light of this, it might be that the trypanocidal effect of Folic acid is not due to Chagas target inhibition, but rather to a beneficial effect of the compound as a supplementary diet. In that sense, the intake of Folic acid might help to improve the immune system of the infected host, indirectly affecting the proliferation of *T. cruzi*. However, further studies are necessary to test this hypothesis.

### 3.4. Drugs Known to Bind Chagas Targets but not Retrieved by the Screening

Two confirmed binders in BindingDB [35] were not predicted by the screening: etravirine, which is an HIV reverse transcriptase inhibitor known to bind cruzipain, and chlorpromazine, an antipsychotic, promiscuous drug that is known to bind the Chagas target trypanothione reductase [91]. A simple explanation for the missing prediction is that a ligand can bind a protein with multiple different binding modes [92], and a PDB structure is only a snapshot of one of these binding modes. For both targets, there were only two query structures available, so it is possible that not all binding modes were captured and therefore the binding mode required to predict the binding of etravirine to cruzipain and chlorpromazine to trypanothione reductase was missing.

### 3.5. Drugs Effective against Trypomastigotes but not against Epimastigotes

During the life cycle of *T. cruzi*, the parasite experiences multiple changes in its morphology, metabolism, and gene expression, going from its epimastigote replicative stage in the insect to its pathogenic metacyclic trypomastigote form [93]. After penetration into the host cell, *T. cruzi* differentiates into the amastigote form and initiates the intracellular binary division. Finally, the amastigotes transform into trypomastigotes, which break open the host cell and enter the bloodstream. The trypomastigotes spread through the bloodstream to penetrate cells of different organs, where the cycle process is repeated [94,95].

The validated drugs in this study were potently effective against the trypomastigote but not the epimastigote form. This behavior can be explained by the morphological differences between the two stages of *T. cruzi* and has already been observed in previous studies [96,97,98]

Over time, it has been assumed that replicating epimastigotes present in the insect gut are not infectious to the mammalian host since only the epimastigote stage is susceptible to the innate immune system of mammals and can be killed by the complement system [99]. However, recent studies have remarked the relevance of treatments that also kill *T. cruzi* epimastigotes [94,100,101].

### 3.6. A Promising Approach for Neglected Diseases

Neglected tropical diseases (NTDs) are a group of bacterial, parasitic, viral, and fungal infections. As they are prevalent in many tropical and sub-tropical developing countries, where poverty is flagrant [102], they are diseases that have not only health but also socio-economic impact. However, they do not receive the attention they require and are generally less studied than other diseases.

Our structure-based drug repositioning approach has proved to be a viable option for the exploration of novel Chagas treatments. Despite being considered a neglected disease, there was sufficient data to conduct a screen for over 75% of targets and return positive results at a very high hit rate, which translates into low cost and time. This is a promising option for all rare and neglected diseases for which reliable treatment has not yet been found.

Structure-based approaches rely on the availability and quality of structural data. Notwithstanding the great amount of structural data available in the PDB, there is an inherent bias towards easily tractable and therapeutically relevant proteins, which is clearly not the case for neglected diseases. Nonetheless, huge efforts are being put into practice to overcome these limitations, e.g., the revolutionary cryogenic electron microscopy technique, which is a more sensitive and widely available approach to solving protein structures [103], or novel modeling techniques that benefit from recent breakthroughs in artificial intelligence [14]. It is expected that such techniques will lead to an immense growth in structural data in the next years and lead to an expansion of promising structural analyses for drug discovery.

## 4. Materials and Methods

### 4.1. Identification and Collection of Chagas Targets and Their Structural Data

Therapeutic Chagas targets were collected from literature evidence. Most of the targets were taken from a preliminary work by Haupt et al. [15] with additional targets added from other literature sources [17,18].

*Selection and characterization of the Chagas targets.* Each target was mapped to the corresponding PDB structures using the UniProt (Universal Protein Resource) accession number. Only targets with a structure available in the PDB were considered for the study. The targets were further characterized and classified according to their Protein family and Enzyme classification categories, primarily using two databases: Pfam (version 31.0) [104] and ENZYME [105]. A protein family is a collection of proteins that have related regions or domains. Knowing about the protein families of the targets provides valuable information about the function of the protein. Furthermore, Enzyme classification was used to analyze whether there are differences between the different enzyme groups in terms of the availability of structural data or the amount of hit compounds.

*Classification and selection of the query ligands.* From the structural point of view, it is important to select high-quality ligands that specifically interact with the target. For this reason, only structures in complex with a ligand were considered. Based on literature and data given in the PDB, the ligands were annotated and divided into 6 different categories: inhibitors, products, substrates, cofactors, fragments, and substances from the buffer or solution used in the crystallization process. In this way, only structures with high quality ligands that specifically interact with their targets (e.g, inhibitors, products, or substrates) were selected for the screening, while cofactors, buffers, and solvents were excluded (Appendix A).

*Sub-classification of binding sites.* The targets were further classified by binding site in order to easily elucidate the predicted binding mechanism of resulting hit compounds. In particular, some of the targets had two different binding/active sites, as shown in Table 1, which was determined via literature review. Additionally, some structures had to be prepared manually with special parameters to detect interaction patterns in alternate atom locations or to split covalently attached cofactors from the ligands (Appendix A).

### 4.2. Computational Screening to Identify Novel Chagas Target Binders

The targets in Table 1 were the input for the *in silico* screening. One independent screening was conducted per target binding site. First, the interaction profiles were obtained from the Protein Ligand Interaction Profiler (PLIP) [106], a tool that detects the non-covalent interactions describing the binding mode of a drug to a target. Second, the interaction profiles were encoded into interaction fingerprints, which are binary vectors. Each bin on the vector represents a feature defined by the combination of two non-covalent interactions within an angle and distance range. This value is set to 1 if the feature is present in the binding or to 0 if it is not, as previously described in Adasme et al. [107]. Finally, each fingerprint was screened against the full PDB to identify other complexes (fingerprints) with a similar binding mode. The results were aggregated by ligands and ranked by *p*-value per screen. That is, if there were multiple complexes of the same drug with different proteins, they were all grouped together and regarded as one hit, with the lowest *p*-value being chosen to represent the group. In addition, the results were filtered to contain only FDA approved drugs or drugs approved elsewhere in the world.

### 4.3. Hit Candidate Prioritization

Following several filtering steps, the primary list of 512 compounds was narrowed to 38 of the top predictions for Chagas disease. First, a *p*-value cut-off of 5 × 10^−4^ was introduced. Subsequently, biologically irrelevant drugs (sugars, amino acids, nucleic acids, fatty acids, citric acids, collagen, glycerol, and others) and compounds with poor drug properties for Chagas (sedatives, disinfectants, hand wash, and others) were excluded, which further decreased the number of hits to 297. Redundant hits were removed while keeping the hit with the lowest *p*-value, reducing the hit count to 164. Each of these hits was evaluated according to three criteria: (1) Visual inspection of binding mode similarity, meaning an interaction pattern similarity with a score greater than 0.75 in case of simple interactions (less than two interaction types or less than five patterns) or greater than 0.6 similarity in case of more complex interactions. (2) Binding affinity values for query protein and hit drug in Binding DB with $IC_{50}$, $K_{d}$, or $K_{i}$ less than or equal to 5 μM. (3) Literature evidence of trypanocidal activity. Direct evidence was defined as an in vitro or in vivo experimental assay in which the drug showed growth inhibition or killing of the parasite. Indirect evidence was defined as the potential of a drug to work against *T. cruzi* (i.e., activity against related organisms, combination therapy studies with known Chagas drugs, etc.) while there is no literature evidence of an assay that directly studies the effect of the drug on *T. cruzi*. If a hit fulfilled at least 1 of the 3 criteria, it was included in the final selection.

### 4.4. Chemical Space of the Hit Candidates.

The chemical similarity of the high priority hits was calculated to assess the novel scaffolds identified in the screen. A chemical similarity score matrix was obtained via PubChem service, where the 2D chemical structure of the compounds is compared pairwise and a similarity score is calculated. The similarity matrix was used as input to generate a clustered heatmap with complete linkage in Rstudio v1.0.153 using the packages heatmap.2 and hclust.

### 4.5. Experimental Validation

The high-priority hit drugs were subjected to experimental validation in vitro, ex vivo, and in vivo, together with a cytotoxicity assay. Ninoa and INC-5 strains of *T. cruzi* were used in this study. The Ninoa strain was obtained in 1986 by xenodiagnoses from an acute case of Chagas’ disease in Oaxaca, Mexico, whereas the INC-5 strain was obtained in 1997 from a chronic case of Chagas’ disease in a 58-year-old woman in Guanajuato, Mexico.

CD1 strain mice, 6–8 weeks old, were used to maintain Ninoa and INC-5 strains of *T. cruzi*. Animals were given by the University of Science and Arts of Chiapas. To maintain the strains, animals were intraperitoneally inoculated every 14 days with 1 × 10^5^ blood trypomastigote, obtained from parasite-infected mice. Parasitemia was monitored every three days by the Pizzi method [108]. For this, 5 µL of blood from the caudal vein of infected mice were deposited on a slide and evenly distributed with an 18 × 18 mm coverslip. Parasite counting was performed by observing 15 microscopic fields using a 40 × objective lens. For the in vivo short term evaluation of the trypanocidal activity of the FDA-approved drugs, six groups of six CD1 mice each with homogeneous weight and parasitemia were used.

*In vitro evaluation of different FDA-approved drugs against Trypanosoma cruzi epimastigotes.* The epimastigote stage of Ninoa and INC5 strains of *T. cruzi* were used to evaluated the in vitro trypanocidal activity of different FDA-approved drugs. Both strains were maintained in liver infusion tryptose (LIT) medium, supplemented with 10% fetal bovine serum (FBS) and 0.1% penicillin–streptomycin. They were preserved by transferring 1 × 10^6^ parasites/mL into a new culture medium every week. The activity of eight FDA-approved drugs as well as nifurtimox and benznidazole—two FDA-approved drugs for Chagas’s disease treatment—were evaluated against *T. cruzi* strains. All compounds were initially prepared at 10 mg/mL, using dimethyl sulfoxide (DMSO) as diluent. Serial dilutions were performed using complemented LIT medium until concentrations of 100 to 0.46 µg/mL of each drug were obtained. *T. cruzi* epimastigotes (1 × 10^6^/well) were cultured in 96-well microliter plates, and incubated for 48 h at 28 °C with the drugs at different concentrations in a final volume of 200 µL. DMSO was included as negative control and nifurtimox and benznidazole as positive controls. After the incubation period, 20 µL of 2.5 mM resazurin solution were added to each well and incubated for 3 h. All assays were carried out in triplicate. The IC_50_ value were determined by Probit analysis [109,110].

*Ex vivo evaluation of different FDA-approved drugs against Trypanosoma cruzi blood trypomastigotes.* The ex vivo evaluation of the trypanocidal activity of the FDA-approved drugs was carried out according to the methodology described above. Blood from parasite infected animals was obtained from intracardiac puncture and diluted with phosphate-buffered saline (PBS), pH 7.2, until a concentration of 2 × 10^6^ blood trypomastigotes/mL was obtained. From this, 195 µL of blood/well was placed in a 96-well plate and incubated at 4 °C for 24 h with 5 µL of each drug tested at different concentrations (5 µg/mL to 250 mg/mL). DMSO was included as negative control and nifurtimox and benznidazole as positive controls. After incubation, live parasites were counted in a Neubauer chamber and the IC_50_ was calculated. Three independent experiments by triplicate were performed.

*In vivo short-term evaluation of the trypanocidal activity of the FDA-approved drugs.* Six groups of six CD1 female mice were infected with blood trypomastigotes of Ninoa and INC-5 strains following the methodology previously described in Romanha et al. [111] and in Díaz-Chiguer et al. [112]. At day 13 post-infection, when the infected mice reached an average of 5 × 10^6^ parasites/mL, a single dose of the FDA-approved drugs at 100 mg/kg body weight was orally administered. Parasitemia was monitored following the Pizzi method at 2, 4, 6, and 8 h post-treatment with a Neubauer chamber using mineral oil as a vehicle. Nifurtimox, vehicle control, and infected non-treated mice were used as controls [108,113]. Statistical significance between the trypanocidal activity of FDA-approved drugs and nifurtimox was analyzed by Student’s t test (Start Plus). Mice examination was performed as stated in the Norma Official Mexicana (NOM-062-Z00-1999), published on August 2009.

*Cytotoxicity assay.* Mouse macrophage cell line J774.2 was cultivated in RPMI medium supplemented with 10% FBS, 100 U µg/mL penicillin, and 100 mg/mL streptomycin, at 37 °C and in an atmosphere of CO_2_ at 5%. The culture medium was replaced at intervals of 2–3 days, according to cell confluence. In order to evaluate the cytotoxicity of the FDA-approved drugs, 50,000 cells/well were placed in a 96-well plate and allowed to adhere for 24 h at 37 °C. Afterwards, FDA-approved drugs were added at 0.8 to 100 µg/mL in a final volume of 200 µL and plates were incubated for 48 h at 37 °C with CO_2_ at 5%. DMSO at 0.1% (the maximum concentration used) was included as negative control, and nifurtimox and benznidazole were used as positive controls. The metabolic activity of the cells was determined following the resazurin method. The percentage of cell viability was calculated and the half maximal cytotoxicity concentration (CC_50_) was determined by Probit analysis. Three independent tests were carried out in triplicates. The selectivity index (SI) was calculated with the formula: CC_50_/IC_50_.

## 5. Conclusions

Chagas disease affects millions of people in South America. As current medications are ineffective and have severe side effects, drug repositioning is a promising approach to find new therapies. The approach presented in this study has demonstrated the potential of structure-based drug repositioning to produce a high rate of true positive hits, which translates into low cost and time. Half of the predicted drugs were already known to have trypanocidal activity, which serves as a good validation of the approach. In addition, the screening proposed a Chagas target for the drugs with no clear mechanism of action, the explanation being based on the similarity of the non-covalent interactions. On the other hand, the screening also predicted drugs with no prior evidence of an association with Chagas disease and with novel chemical scaffolds, indicating the potential of the approach to reveal promising drug repositioning candidates. We present three candidate Chagas drugs: ciprofloxacin, naproxen, and folic acid, which demonstrated inhibitory activity against *T. cruzi* trypomastigotes ex vivo and in vivo. Importantly, these candidates are inexpensive, easily available, widely used, and have few side effects. This makes them perfect candidates for further validation in models and in humans, and could eventually lead to a more efficient and better tolerated treatment of Chagas disease. Overall, our approach represents a promising option for all rare and neglected diseases for which no reliable treatment has yet been found.

## Figures and Tables

**Figure 1 ijms-21-08809-f001:**
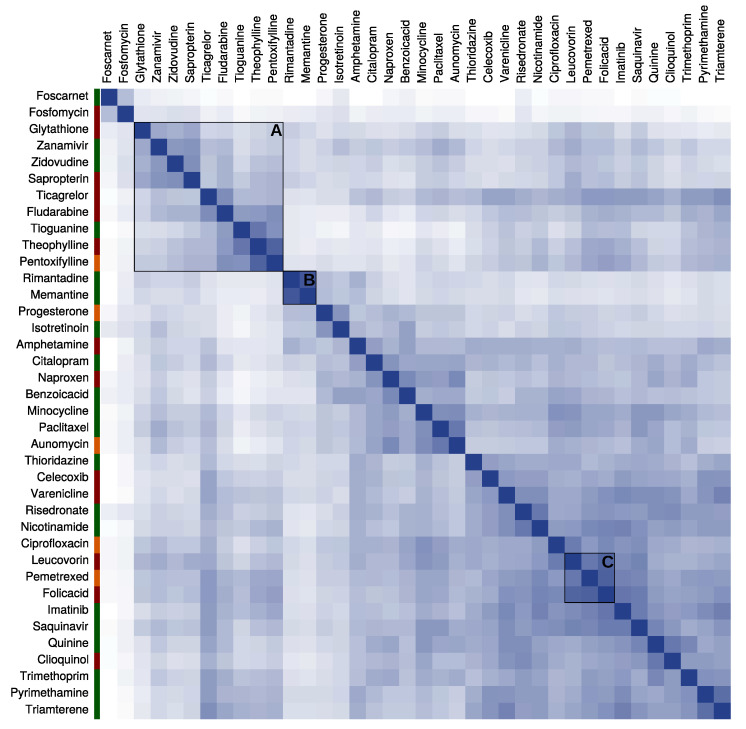
Chemical space of the hit candidates. The heatmap shows the pairwise similarity of the chemical structures of the hits. The similarity scores range from 0 (low) to 1 (high) with a color scheme from white to blue, respectively. The color tags on the right indicate the novelty of the drug with regard to the trypanocidal activity: direct evidence (green), indirect evidence (orange), and no previous evidence (red). Some clusters of drugs with a relatively high chemical similarity are marked: (**A**) nucleic acid analogs, (**B**) adamantane analogs, and (**C**) folate analogs.

**Figure 2 ijms-21-08809-f002:**
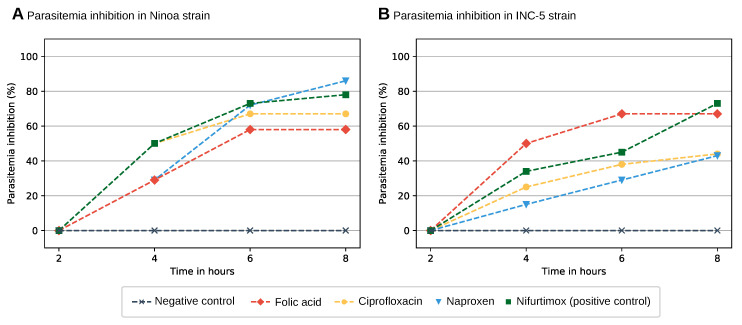
Parasitemia inhibition (%) of *T. cruzi* NINOA (**A**) and INC-5 (**B**) strains by the tested FDA-approved drugs during 8 h after administration. The plot shows parasitemia inhibition by the drugs ciprofloxacin (yellow line), naproxen (blue line), and folic acid (red line) at 2, 4, 6, and 8 h after administration. Drugs at a single dose of 100 mg/kg body weight were orally administered at day 13 post-infection when the infected mice reached an average parasitemia of 5 × 10^6^ parasitemia/mL. Infected mice treated with nifurtimox (green line) and infected non-treated mice (gray line) were used as positive and negative controls, respectively.

**Figure 3 ijms-21-08809-f003:**
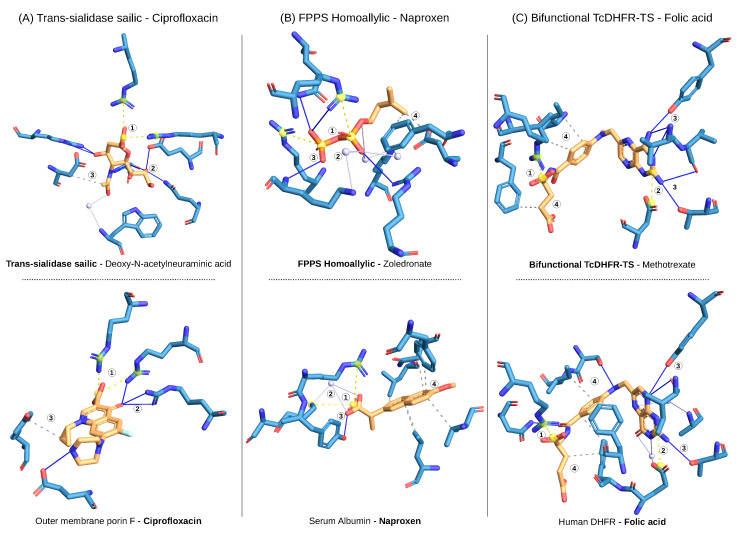
Non-covalent interaction patterns accounting for the repositioning predictions. The structure-based drug repositioning approach predicted that ciprofloxacin binds trans-sialidase (sialic acid site), naproxen binds FPPS (homoallylic site) binder, and folic acid binds TcDHFR. The repositioning is based on the similarity of the non-covalent interactions defining the binding mode of inhibitors (orange) to their targets (blue) between query (top) and hit (bottom) complexes. (**A**) The binding mode of ciprofloxacin to porin F (PDB ID: 4kra) is similar to the one of deoxy-N-acetylneuraminic acid to trans-sialidase (sialic acid site) (PDB ID: 1ms0). Both have in common (1) a double salt bridge (yellow dashed lines), (2) a triple set of hydrogen bonds (blue lines), and (3) a hydrophobic interaction (gray dashed lines). In the same way, (**B**) the binding mode of naproxen to serum albumin (PDB ID: 4ot2) is similar to the one of zoledronate to FPPS (PDB ID: 3iba) as they have (1) two salt bridges, (2) one water bridge (lightblue lines and sphere), and (3) one hydrogen bond in common. (**C**) The binding mode of folic acid to human DHFR (PDB ID: 1drf) is similar to the one of methotrexate to TcDHFR (PDB ID: 3cl9) with (1,2) two salt bridges, (3) a set of hydrogen bonds, and (4) a set of hydrophobic interactions.

**Table 1 ijms-21-08809-t001:** *T. cruzi* targets used as input for the computational screening. The targets above the bold line have been thoroughly researched and there is a high confidence that modulating these targets will produce the desired effect in *T. cruzi*. The targets below the line have been at least researched for being involved in *T. cruzi*’s survival. For each target, the target name, the specific binding site (in case of a multi-binding site target), the enzyme class, the PDB ID of the available structures, and the binding ligand are indicated. An independent screening was conducted for each target binding site. The last column on the right indicates the number of hit complexes predicted for each screening.

	Target	Binding Site	Enzyme Class	PDB ID	Ligand Name	Ligand Type	Nr. Hits
1	Bifunctional DHFR-TS [19]	DHFR site	Oxidoreductase	3IRN	Cycloguanil	Inhibitor	43
				3CLB	Trimetrexate	Inhibitor	
				3CL9	Methotrexate	Inhibitor	
				3HBB	Trimetrexate	Inhibitor	
				3IRM	Cycloguanil	Inhibitor	
		TS site	Transferase	3CL9	DUMP	substrate	7
2	Cruzipain [20]	unspecified	Transferase	2OZ2	K11777	Inhibitor	15
				3LXS	WRR483	Inhibitor	
3	FPPS [21]	allylic site	Transferase	3IBA	Zoledronate	Inhibitor	20
				1YHL	Risedronate	Inhibitor	
				1YHM	Alendronate	Inhibitor	
				3ICK	Minodronate	Inhibitor	
		homoallylic site	Transferase	1YHM	Isopentyl Pyrophosphate	Substrate	8
				3ICK	Isopentyl Pyrophosphate	Substrate	
				3IBA	Isopentyl Pyrophosphate	Substrate	
4	GAPDH [22]	active site	Oxidoreductase	1QXS	1,3-bisphospho-d-glyceric acid	Product	18
				1K3T	Chalepin	Inhibitor	
		covalent site	Oxidoreductase	3IDS	Iodoacetamide	Inhibitor	24
5	Lanosterol Demethylase [23]	unspecified	Oxidoreductase	5AJR	VT-1161	Inhibitor	69
				3ZG3	UDD	Inhibitor	
				3ZG2	UDO	Inhibitor	
				3KSW	VNF	Inhibitor	
				2WX2	Fluconazole	Inhibitor	
				3KHM	Fluconazole	Inhibitor	
				2WUZ	Fluconazole	Inhibitor	
				3K1O	Posaconazole	Inhibitor	
6	trans-sialidase [24]	acceptor site	Hydrolase	1MS9	Beta-lactose	Substrate	20
				1MS0	Beta-lactose	Substrate	
		sialic acid site	Hydrolase	1MS0	DANA	Inhibitor	12
				1S0J	Methylumbelliferyl Sialic Acid	Substrate	
				1S0I	Lactose Sialic Acid	Substrate	
7	Trypanothione Reductase [25]	unspecified	Oxidoreductase	1GXF	Quinacrine Mustard	Inhibitor	13
				1BZL	Trypanothione	Substrate	
8	B Cell Mitogen [24]	unspecified	Isomerase	1W61	Pyrrole-2-Carboxylic acid	Substrate	10
9	Dihydroorate Dehydrogenase [26]	unspecified	Oxidoreductase	2E6A	Orotate	Product	16
				2E6D	Fumarate	Substrate	
				2E68	Dihydroorotate	Substrate	
				2DJL	Succinate	Product	
				2E6F	Oxonate	Inhibitor	
10	Glucose 6-phosphate dehydrogenase [18]	unspecified	Oxidoreductase	6D24	Beta-Glucose-6 Phosphate	Substrate	2
				5AQ1	Beta-Glucose-6 Phosphate	Substrate	
11	HGPRT [27]	PRPP site	Transferase	1TC2	PRPP	Substrate	0
		purine site	Transferase	1TC2	7HPP	Inhibitor	42
				1P19	Inosinic Acid	Product	
				1TC1	Formycin B	Product	
12	Old Yellow Enzyme [28]	unspecified	Oxidoreductase	3ATZ	Hydroxybenzaldehyde	Substrate	34
13	Pteridine Reductase [29]	unspecified	Oxidoreductase	1MXF	Methotrexate	Inhibitor	26
				1MXH	Dihydrofolic Acid	Substrate	
14	Spermidine Synthase [30]	dcSAM site	Transferase	5B1S	dcSAM	Substrate	6
				4YUW	dcSAM	Substrate	
		putrescine site	Transferase	5B1S	2-(2-fluorophenyl)ethanamine	Inhibitor	30
				4YUW	trans-4-methylcyclohexanamine	Inhibitor	
15	Squalene Synthase [17]	unspecified	Transferase	3WSB	SQ109	Inhibitor	97
				3WCA	Farnesyl Thiopyrophosphate	Substrate	
				3WCB	BPH1237	Inhibitor	
				3WCC	E5700	Inhibitor	
				3WCE	ER119884	Inhibitor	
				3WCG	BPH1344	Inhibitor	
16	UDP-galactapyranose mutase [31]	unspecified	Isomerase	4DSH	UDP	Substrate	11
				4DSG	UDP	Substrate	

**Table 2 ijms-21-08809-t002:** Computational screening hits predicted to bind Chagas targets. Top 38 hits from the screening sorted by novelty with regard to trypanocidal activity, where ? indicates unknown activity, + indicates indirect evidence of activity, and ++ indicates direct evidence of activity. Within each of the previous classifications, the hits are sorted by screening *p*-value and the original indication is shown. The ✔ represents positive visual inspection, meaning high interaction pattern similarity between query and hit.

	Drug	Predicted Target	*p*-Value	Current Indication	Visual Inspection	Trypan. Activity	Reference
1	Glutathione	Dihydroorate	3.80 × 10^−6^	Antioxidant	✔	?	
2	Naproxen	FPPS Homoallylic	6.46 × 10^−6^	Anti-inflammatory	✔	?	
3	Amphetamine	Spermidine Synthase Putrescine	1.14 × 10^−5^	Attention deficit/Hyperactivity	✔	?	
4	Folic acid	DHFR	2.66 × 10^−5^	Megaloblastic Anemia	✔	?	
5	Sapropterin	Pteridine Reductase	2.66 × 10^−5^	Phenylketonuria	✔	?	
6	Clioquinol	Trypanothione Reductase	2.66 × 10^−5^	antifungal	✔	?	
7	Celecoxib	DHFR	3.04 × 10^−5^	Anti-inflammatory	✔	?	
8	Leucovorin	DHFR	4.18 × 10^−5^	Toxicity of Pyrimethamine	✔	?	
9	Theophylline	HGPRT purine	7.22 × 10^−5^	Asthma	✔	?	
10	Fosfomycin	FPPS allylic	1.52 × 10^−4^	Antibiotic	✔	?	
11	Ticagrelor	Pteridine Reductase	1.90 × 10^−4^	Platelet aggregation inhibitor	✔	?	
12	Fludarabine	HGPT purine	4.18 × 10^−4^	Cancer	✔	?	
13	Varenicline	Spermidine Synthase Putrescine	4.45 × 10^−4^	Nicotine Addiction	✔	?	
14	Progesterone	Old Yellow	3.80 × 10^−6^	Hormone	✔	+	Schuster et al. [36]
15	Pemetrexed	Pteridine Reductase	3.80 × 10^−6^	Cancer	✔	+	Sienkiewicz et al. [37]
16	Ciprofloxacin	Trans-sialidase sailic	9.51 × 10^−5^	Antibiotic	✔	+	Hiltensperger et al. [38]
17	Aunomycin	Trypanothione Reductase	1.29 × 10^−4^	Cancer	✔	+	Andrews et al. [39]
18	Pentoxifylline	Trypanothione Reductase	2.24 × 10^−4^	Muscle Pain Reliever	✔	+	Villa-Pereira et al. [40]
19	Pyrimethamine	DHFR	3.80 × 10^−6^	Toxoplasmosis	✔	++	Gilbert et al. [19]
20	Trimethoprim	DHFR	3.80 × 10^−6^	Antibiotic	✔	++	Gilbert et al. [19]
21	Risedronate	FPPS Allylic	3.80 × 10^−6^	Osteoporosis	✔	++	Huang et al. [41]
22	Triamterene	Pteridine Reductase	3.80 × 10^−6^	Diuretic	✔	++	Planer et al. [13]
23	Tioguanine	Pteridine Reductase	7.61 × 10^−6^	Cancer	✔	++	fernandes et al. [42]
24	Nicotinamide	Spermidine Synthase Putrescine	7.61 × 10^−6^	Pellagra	✔	++	Soares et al. [43]
25	Zidovudine	Squalene Synthase	1.14 × 10^−5^	HIV	✔	++	Nakajima-Shimada et al. [44]
26	Zanamivir	Trans-sialidase sailic	1.14 × 10^−5^	Antiviral	✔	++	Kashif et al. [45]
27	Rimantadine	DHFR	1.54 × 10^−5^	Anti-viral	✔	++	Kelly et al. [46]
28	Quinine	DHFR	1.90 × 10^−5^	Malaria	✔	++	Ceole et al. [47]
29	Benzoic acid	Galactopyranose Mutase	1.90 × 10^−5^	Antifungal		++	Neres et al. [48]
30	Imatinib	Cruzipain	6.84 × 10^−5^	Cancer	✔	++	Simoes-Silva et al. [49]
31	Isotretinoin	DHFR	7.22 × 10^−5^	Acne	✔	++	Reigada et al. [50]
32	Foscarnet	FPPS allylic	7.61 × 10^−5^	Antiviral	✔	++	Haupt et al. [15]
33	Paclitaxel	Trans-sailidase Acceptor	8.75 × 10^−5^	Cancer		++	Baum et al. [51]
34	Citalopram	Squalene Synthase	9.13 × 10^−5^	Antidepressant	✔	++	Jones et al. [52]
35	Thioridazine	Squalene Synthase	1.48 × 10^−4^	Antipsychotic	✔	++	Lo Presti et al. [53]
36	Memantine	Squalene Synthase	2.51 × 10^−4^	Alzheimer’s	✔	++	Damasceno et al. [54]
37	Saquinavir	Spermidine Synthase dcSAM	2.70 × 10^−4^	HIV	✔	++	Sangenito et al. [55]
38	Minocycline	Spermidine Synthase Putrescine	3.65 × 10^−4^	Antibiotic		++	Planer et al. [13]

**Table 3 ijms-21-08809-t003:** Inhibition of proliferation of Ninoa and INC-5 strains of *T. cruzi*, cytotoxicity and selectivity index of tested FDA-approved drugs compared to the known treatments with nifurtimox and benznidazole (positive controls).

	*T.cruzi* blood trypomastigote		Cytotoxicity	Selectivity Index		*T.cruzi* epimastigote		Cytotoxicity	Selectivity Index	
Drug	IC_50_ (µM)		CC_50_ (µM)	CC_50_/IC_50_		IC_50_ (µM)		CC_50_ (µM)	CC_50_/IC_50_	
	Ninoa	INC-5		Ninoa	INC-5	Ninoa	INC-5		Ninoa	INC-5
**Ciprofloxacin**	**25.7 ± 0.04**	**21.3 ± 0.09**	**7.7** × **10^23^ ± 0.31**	**3.0** × **10^22^**	**3.6** × **10^22^**	**> 400 ± 0.17**	**> 400 ± 0.17**	**7.7** × **10^23^ ± 0.31**	**< 1.9** × **10^21^**	**< 1.9** × **10^21^**
**Folic Acid**	**28.1 ± 0.01**	**21.5 ± 0.03**	**9.1** × **10^17^ ± 0.31**	**3.2** × **10^16^**	**4.2** × **10^16^**	**> 400 ± 0.03**	**> 400 ± 0.01**	**9.1** × **10^17^ ± 0.14**	**< 2.3** × **10^15^**	**< 2.3** × **10^15^**
**Naproxen**	**58.5 ± 0.07**	**38.3 ± 0.06**	**2.5** × **10^18^ ± 0.16**	**4.3** × **10^16^**	**6.5** × **10^16^**	**> 400 ± 0.10**	**> 400 ± 0.04**	**2.5** × **10^18^ ± 0.08**	**< 6.3** × **10^15^**	**< 6.3** × **10^15^**
Celecoxib	> 100 ± 0.06	> 100 ± 0.15	1.2 × 10^16^ ± 0.19	< 1.2 × 10^14^	< 1.2 × 10^14^	> 400 ± 0.13	> 400 ± 0.02	1.2 × 10^16^ ± 0.32	< 3.1 × 10^15^	< 3.1 × 10^15^
Glutathione	> 100 ± 0.08	> 100 ± 0.05	1.7 × 10^18^ ± 0.07	< 1.7 × 10^14^	< 1.7 × 10^14^	> 400 ± 0.10	> 400 ± 0.17	1.7 × 10^18^ ± 0.42	< 4.3 × 10^15^	< 4.3 × 10^15^
Leucovirin	> 100 ± 0.18	> 100 ± 0.05	7.9 × 10^23^ ± 0.23	< 7.9 × 10^21^	< 7.9 × 10^21^	> 400 ± 0.12	> 400 ± 0.08	7.9 × 10^23^ ± 0.17	< 2.0 × 10^21^	< 2.0 × 10^21^
Pentoxyfiline	> 100 ± 0.25	> 100 ± 0.12	7.9 × 10^17^ ± 0.05	< 7.9 × 10^15^	< 7.9 × 10^15^	> 400 ± 0.14	> 400 ± 0.16	7.9 × 10^17^ ± 0.02	< 2.0 × 10^15^	< 2.0 × 10^15^
Theophyline	> 100 ± 0.18	> 100 ± 0.15	8.3 × 10^35^ ± 0.14	< 8.3 × 10^33^	< 8.3 × 10^33^	> 400 ± 0.21	> 400 ± 0.03	8.3 × 10^35^ ± 0.08	< 2.1 × 10^33^	< 2.1 × 10^33^
Nifurtimox	167.1 ± 0.03	115.2 ± 0.17	164.2 ± 0.25	0.10	1.42	7.09 ± 0.12	6.47 ± 0.42	164.2 ± 0.08	23.2	25.4
Benznidazole	156.0 ± 0.11	130.6 ± 0.08	133.9 ± 0.06	0.85	1.02	30.3 ± 0.03	19.9 ± 0.23	133.9 ± 0.71	4.42	6.74

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
