# Peer review of "Repositioned Drugs for Chagas Disease Unveiled via Structure-Based Drug Repositioning"

_ijms, 2020, doi:10.3390/ijms21228809_

Round 1
Reviewer 1 Report
Because the pharmaceutical industry has little financial interest in the developing new drugs for the neglected tropical diseases, the authors demonstrated drug repositioning (off label use or new indications for already approved drugs) to find ciprofloxacin, naproxen, and folic acid, showed a growth inhibitory activity in the micromolar range when tested ex vivo on T. cruzi trypomastigotes.
I have some commends:
- The method is helpful for developing new drugs (indications) for the neglected tropical diseases.
- In vivo test (figure 2), were the 3 candidates (ciprofloxacin, naproxen, and folic acid) significantly different from the positive (nifurtimox) control?
- The dose was very high (100 mg/kg body weight). For a person with 60 kg body weight, he/she needs 6 g of ciprofloxacin, naproxen, or folic acid. It should discuss.
Reviewer 2 Report
Chagasa disease is one of the most dangerous Neglected Tropical Diseases. According WHO about 7 million people worldwide are infected with Trypanosoma cruzi, the parasite that causes Chagas disease.
The two drugs (Benznidazole, Nifurtimox) currently available on the marked and used for treatment feature numerous side effects - they produce severe adverse effects in patients. Therefore, the need to develop new therapies, efficient treatment methods, including finding suitable drugs (specific, highly efficient, without side effects/less toxic, inexpensive, easily accessible). A good solution may be the testing of the efficacy of the already approved or experimental antiparasitic drugs.
The manuscript authors selected three (ciprofloxacin, naproxen, folic acid) new substances/candidates against Chagasa disease, which exhibited inhibiting effect on T. cruzi trypomastigotes. At the same time, they are inexpensive, accessible and produce few side effects.
I recommend this paper for publication in the International Journal of Molecular Sciences after minor revisions.
Comments/questions:
- Line 438: were the blood smears stained ?,
- Line 441: is the sex of the experimental mice known ?; is it possible that host sex may be significant here (affect the parasitemia ?).
- References: entry not compliant with IJMS (e.g. journal abbreviation, titles written in different fonts (upper and lowercase letters), please correct.
